# *MdMYB66* Is Associated with Anthocyanin Biosynthesis via the Activation of the *MdF3H* Promoter in the Fruit Skin of an Apple Bud Mutant

**DOI:** 10.3390/ijms242316871

**Published:** 2023-11-28

**Authors:** Yaping Huang, Wenfang Li, Shuzhen Jiao, Juanjuan Huang, Baihong Chen

**Affiliations:** 1College of Horticulture, Gansu Agricultural University, Lanzhou 730070, China; hyaping0101@163.com (Y.H.); liwenf@gsau.edu.cn (W.L.); shuzhenjiao@126.com (S.J.); 18719853694@163.com (J.H.); 2Tianshui Institute of Pomology, Tianshui 741002, China

**Keywords:** apple, mutant, anthocyanin biosynthesis, *MdMYB66* transcription factor

## Abstract

Skin color is an important trait that is mainly determined by the content and composition of anthocyanins in apples. In this study, a new bud mutant (RM) from ‘Oregon Spur II’ (OS) of Red Delicious apple was obtained to reveal the mechanism underlying red color formation. Results showed that the total anthocyanin content in RM was significantly higher than that in OS with the development of fruit. Through widely-targeted metabolomics, we found that cyanidin-3-O-galactoside was significantly accumulated in the fruit skin of RM. Transcriptome analysis revealed that the structural gene *MdF3H* and *MdMYB66* transcription factor were significantly up-regulated in the mutant. Overexpression of *MdMYB66* in apple fruit and apple callus significantly promoted anthocyanin accumulation and significantly increased the expression level of *MdMYB66* and structural genes related to anthocyanin synthesis. Y1H and LUC analysis verified that *MdMYB66* could specifically bind to the promoter of *MdF3H*. The results of the double luciferase activity test showed that *MdMYB66* activated *MdF3H* 3.8 times, which led to increased anthocyanin contents. This might explain the phenotype of red color in RM at the early stage. Taken together, these results suggested that *MdMYB66* was involved in regulating the anthocyanin metabolic pathways through precise regulation of gene expression. The functional characterization of *MdMYB66* provides insight into the biosynthesis and regulation of anthocyanins.

## 1. Introduction

Fruits play an important role in people’s daily diets and supply various nutrients needed in the human body. The quality and nutritional value of fruits are the key factors influencing consumer choice and acceptance [1]. In plants, the main pigments for fruit coloring are flavonoids/anthocyanins, carotenoids, and chlorophyll [2]. According to their structures, flavonoids include flavonols, chalcones, anthocyanins, and flavones [3]. Anthocyanins are a class of flavonoids and water-soluble pigments that endow plant organs and tissues with a variety of colors, such as red, purple, or blue. The patterns and quantities of anthocyanins are the main factors influencing fruit quality [2]. Most important of all, anthocyanin-rich fruits are favored by people because of their high nutritional value for human health. For example, they have been used to treat conditions as diverse as hypertension, pyrexia, liver disorders, the common cold, and so on. They have even been found to improve vision and blood circulation [4]. Therefore, the development of fruits with high anthocyanin content has been a major goal for breeders and researchers because of the health benefits of these pigments.

Apple (*Malus domestica* Borkh.) is a very popular and economical fruit all over the world due to its high anthocyanin content, nutritional value, and various medical properties. Moreover, apples contain lower calories and higher mineral, pectin, and vitamin content and are favored by people [5,6]. Skin color, which contains high anthocyanins, is one of the most important traits affecting the appearance and quality of fruits. The composition and concentration of anthocyanins in the skin are responsible for the different color phenotypes. The apple cultivar ‘Red Delicious’ is the most susceptible to bud mutation, allowing researchers to screen for fruits with excellent traits, such as good color and flavor, through phenotypic observation. 

The most representative metabolic pathway of anthocyanin biosynthesis is phenylpropanoid and flavonoid pathways, which are catalyzed by a series of enzymes [7]. Phenylalanine ammonia-lyase (PAL), cinnamate 4-hydroxylase (C4H), and 4-coumarate CoA ligase (4CL) are the main enzymes involved in the phenylpropanoid pathway [8]. Chalcone synthase (CHS), chalcone isomerase (CHI), flavone 3-hydroxylase (F3H), flavonoid 3′-hydroxylase (F3′H), and flavanone 3′5′-hydroxylase (F3′5′H) are the important catalytic enzymes involved in flavonoid biosynthesis. Anthocyanidins are synthesized by dihydroflavonol-4-reductase (DFR), anthocyanidin synthase (ANS), and UDP-glucose/flavonoid 3-O-glucosyltransferase (UFGT) [9]. Most of these structural genes have been found to be involved in the color formation of fruit skin. ANS is an important enzyme at the end of the anthocyanin biosynthesis in plants, which catalyzes the transition from colorless to colored anthocyanins [10]. UFGT is also the key enzyme involved in anthocyanin synthesis, and the activities of other enzymes, such as PAL and DFR, are not correlated with anthocyanin synthesis in apple skin [11]. With the accumulation of anthocyanin, the expression levels of *F3H*, *ANS*, and *UFGT* related to anthocyanin biosynthesis were up-regulated in ‘Jonathan’ and ‘Fuji’ apple skins [12]. Furthermore, the transcription factors (TFs) of MYB, bHLH, WD40, and MYB-bHLH-WD40 (MBW) protein complex can also modulate anthocyanin biosynthesis. The important roles of these transcription factors in anthocyanin synthesis have been widely studied in plants [13]. In apple skin, the transcription factors *MdMYB1* and *MdMYBA* were initially identified as regulating anthocyanin biosynthesis. Subsequently, many other MYB TFs, including *MdMYB10*, *MdMYB11*, *MdMYB12*, and *MdMYB22*, have also been isolated in regulating anthocyanin biosynthesis and flavanol and flavanol biosynthesis [14]. In addition to MYB TFs, bHLH TFs have also been shown to regulate anthocyanin biosynthesis in apple fruit. For example, *bHLH3* can bind to the promoters of *DFR* and *UFGT* to increase anthocyanin synthesis [15]. *MdTTG1* belonging to WD40 may regulate anthocyanins by interacting with *MdbHLH3* and *MdbHLH33* [16].

With the further development of omics, such as transcriptomics, metabolomics, and proteomics, integrated analysis of the two omics, or even more, has been extensively used to elucidate regulatory genes and metabolic pathways in horticultural plants, especially in fruit coloration [17]. Moreover, the regulatory mechanisms of anthocyanin biosynthesis are rather complex in horticultural crops, which facilitates carrying out further research on fruit coloration. In this study, a new red bud mutant from ‘Oregon Spur Ⅱ’ of Red Delicious apples was discovered in an apple orchard in Tianshui County of Gansu Province, China. Interestingly, the fruit skin of mutants appears deep red at the young fruitlet stage. The difference in fruit skin coloration between ‘Oregon Spur Ⅱ’ and its new red mutant is, as yet, unclear. Generally, the young fruitlet stage and the mature fruit stage are two important stages of anthocyanin accumulation in fruit [12]. Regarding the molecular mechanism of anthocyanin biosynthesis, previous studies have mainly focused on the fruit ripening stage, which was thought to be the important period of color formation. Herein, to understand the coloring mechanism of ‘Oregon Spur Ⅱ’ and its red mutant at different developmental stages, transcriptomic integrated with metabolomic was performed, a key regulatory gene, *MdMYB66*, was uncovered as a novel R2R3-type MYB transcription factor, and the mechanism of its regulation of anthocyanin biosynthesis was discussed. The results could provide essential theoretical support for developing bioactive compounds and cultivating anthocyanin-rich red apple varieties.

## 2. Results

### 2.1. Color Phenotypes and Anthocyanin Contents in OS and RM at Different Developmental Stages

Six time points, including 20, 60, 100, 120, 130, and 140 DAF (days after fluorescence), were analyzed for the dynamic changes of total anthocyanin contents in RM and OS. As seen from Figure 1B, the color phenotypes of fruits continued to increase over time by visual inspection. The fruit skin color of OS kept green at 20 and 60 DAF until 100 DAF, when it gradually began coloring, while the fruit skin color of RM was completely red from 20 to 140 DAF. As shown in Figure 1C, during the overall developmental stage, the total anthocyanin content in RM was significantly higher than that in OS. In particular, the anthocyanin contents increased rapidly at 130 and 140 DAF, with values of 1.32 mg/g and 2.55 mg/g in RM, respectively. In general, the differences in total anthocyanin contents were obvious between OS and RM, and each stage showed a significant increase in RM, indicating that the red coloration of RM fruit skin was associated with increased anthocyanin contents.

### 2.2. Variations in Anthocyanin Composition and Concentration in OS and RM 

A total of 25 anthocyanin components were obtained through widely targeted anthocyanin metabolomics analysis between OS and RM at the S1 (20 DAF) and S2 (130 DAF) stages (Appendix A). Principal component analysis (PCA) and OPLS-DA results showed that the three replicates of OS and RM were concentrated in the score chart, indicating that the differences within each treatment were small and that the stability of this test was good (Appendix A). To compare the overall difference of anthocyanins, the relative abundance of each anthocyanin was calculated based on the mass spectrum signal intensity (Figure 2A,B). Obviously, the concentration of cyanidin-3-O-glucoside, cyanidin-3-O-galactoside, and cyanidin-3-O-xyloside were significantly higher, whereas the accumulation levels of procyanidin C1 and procyanidin B1 were significantly lower in RM than that in OS at S1 and S2 stage. There was no significant change in procyanidin B2. As expected, the red skin of RM responsible for deep red pigmentation was due to its high levels of cyanidin compounds. To the contrary, significantly lower levels of cyanidin anthocyanins were detected in OS. Meanwhile, based on the relative abundance of anthocyanins, hierarchical cluster heatmap analysis results showed that the relative abundance of five anthocyanins, including cyanidin-3-O-galactoside, cyanidin-3-O-arabinoside, cyanidin-3-O-xyloside, peonidin-3-O-glucoside, and cyanidin-3-(6-O-p-caffeoyl)-glucoside were up-regulated in RM at the S1 stage (Figure 2C). However, in addition to the above five anthocyanins, peonidin-3-O-rutinoside, peonidin-3-O-arabinoside, peonidin-3-O-glucoside, pelargonidin-3-O-galactoside, procyanidin A2 and kaempferol-3-O-rutinoside were also up-regulated at the S2 stage (Figure 2D). Therefore, these results suggested that the pigment Cy (cyanidin-3-O-glucoside, cyanidin-3-O-galactoside, and cyanidin-3-O-xyloside) was the key anthocyanin metabolites contributing to the change in skin coloration of RM during early developmental period. However, during the ripening period, the pigments Cy and Pn (peonidin-3-O-rutinoside, peonidin-3-O-arabinoside, peonidin-3-O-glucoside) were the common anthocyanin metabolites in skin coloration of RM compared with OS. Furthermore, the concentration of procyanidin, such as procyanidin B1, procyanidin B2, and procyanidin C1, was detected at a high level in OS during all the stages, indicating that this may be another reason for the color difference between OS and RM. 

### 2.3. Identification of the Differentially Accumulated Anthocyanins (DAAs) in OS and RM

The anthocyanins of OS and RM accumulated at 20 DAF and 130 DAF were compared. A total of 14 anthocyanin components were significantly and differentially expressed at 20 DAF; the number of up- and down-regulated were 3 and 11, respectively. In addition, 18 anthocyanin components were significantly and differentially expressed at 130 DAF, and the number of up- and down-regulated were 12 and 6 between OS and RM, respectively (Figure 3A and Appendix A). Subsequently, in order to assess the impact of the metabolite difference between the fruit skin color, FC (fold change) ≥ 1.5 was used to screen some important metabolites by combining the *p*-value (*p* < 0.05). These important metabolites are clearly shown in Figure 3B,C. In total, 4 and 9 DAAs were significantly presented between OS and RM and varied greatly at 20 DAF and 130 DAF. Therefore, the results suggested that these four DAAs, including cyanidin-3-O-galactoside, cyanidin-3-O-glucoside, cyanidin-3-O-xyloside, and cyanidin-3-O-arabinoside were main pigments at the early developmental stage in OS and RM. However, at the maturity stage, delphinidin, kaempferol, pelargonidin, peonidin, and procyanidin A2 were also significantly increased, indicating that the coloring of fruit was the result of the cooperative effect of various pigments.

### 2.4. Transcriptome Profiles of OS and RM

To investigate the difference in gene expression level in anthocyanin biosynthesis of OS and RM, transcriptomic analysis was performed at the 20 DAF stage. A total of 34.80 GB of clean data were generated with an average of 5.80 GB per sample, and the rate of total mapping ranged from 87.52% to 89.51% (Table 1). Based on the number of counts per million (CPM) values, the PCA of the two samples showed that the three biological replicates of each sample were distributed together, indicating that the transcriptomic sequencing data were accurate for further study. The PC1 accounted for most of the change in the data, with a value of 60.1% (Figure 4A). The PCA result of the transcriptome was consistent with the metabolome analysis, and there was an obvious separation between OS and RM. The result suggested that the differences between gene expression and metabolite accumulation were correlative during the fruit developmental stage.

To identify the DEGs in OS and RM, a transcriptomic comparison was performed. Based on Fold Change ≥ 1.5 and FDR < 0.05 criteria, a total of 843 DEGs, including 289 up-regulated and 554 down-regulated, were found differentially expressed between OS and RM (Figure 4B). GO function and KEGG pathway enrichment were analyzed. GO enrichment top 50 analysis of the 843 DEGs showed that there were three processes. In the biological process, most of the DEGs were enriched in the cellular process and in the metabolic process. In the cellular component process, most of the DEGs were enriched in cellular anatomical entities. In the molecular function process, the DEGs were enriched in binding and catalytic activity. In addition, KEGG pathway enrichment analysis results showed that the DEGs were mainly enriched in metabolic pathways, followed by Environmental information processing pathways. Furthermore, two metabolic pathways of phenylpropanoid biosynthesis and flavonoid biosynthesis were related to anthocyanin biosynthesis (Figure 4C,D).

### 2.5. Structural Genes and Transcription Factors Related to Anthocyanin Synthesis in OS and RM 

To reveal the mechanism underlying skin color formation in OS and RM, the key genes related to flavonoid biosynthesis and phenylpropanoid biosynthesis were selected from the transcriptional database. Among these secondary metabolic pathways, six differentially expressed structural genes were obtained. Notably, the structural genes of two *4CL* in the upstream pathway and the *F3H* related to anthocyanin synthesis were up-regulated, while the genes of *CCR*, *CAD*, and *POD* related to lignin synthesis were down-regulated in RM compared with OS (Figure 5A). Previous studies suggested that MYB and bHLH TF are the key transcription factors that regulate anthocyanin biosynthesis. In this study, a total of 7361 transcription factors were predicted from the obtained new transcripts, including 275 bHLH transcription factors and 229 MYB transcription factors (Appendix A). Six transcription factors related to anthocyanin synthesis (*bHLH96*-*like*, *MYB113-like*, *MYB114-like1*, *MYB114-like2*, *MYB1R1-like*, *MYB66*) were significantly up-regulated in RS. Moreover, correlation analysis of transcriptomics and metabolomics showed that two differentially expressed genes, including *F3H* and *MYB66*, were positively correlated with cyanidin-based anthocyanin pigments. In contrast, the genes of *CCR*, *CAD*, and *POD* were negatively related to these pigments, which could be the main reason for anthocyanin accumulation and the color difference between OS and RM at the early developmental stage (Figure 5B).

### 2.6. qRT-PCR Validation of DEGs

To further validate the transcriptome data, 12 candidate DEGs were analyzed by qRT-PCR. Among these genes, six structural genes, including two up-regulated *4CL* (LOC103427406, LOC103426517), one up-regulated *F3H* (LOC103436139), one down-regulated *CAD* (LOC103425085), one down-regulated *CCR* (LOC103943280), and one down-regulated *POD* (LOC103450728) were confirmed. In addition, six transcription factors (TFs), including four up-regulated MYB TFs (LOC103421948, LOC103455780, LOC103424517, LOC103431271) and two bHLH TFs (LOC114821749, LOC103433823) were also verified by qRT-PCR. The results showed that the expression profiles of these candidate DEGs were consistent with the transcriptome data, which further demonstrated the credibility of the RNA-Seq data generated in this study (Figure 6).

### 2.7. Isolation and Analysis of MdMYB66 Transcription Factor

The transcript (MD14G1181000) was extremely up-regulated, and correlation analysis showed a positive correlation with cyanidin-based anthocyanin pigments at 20 DAF period in the mutant. The sequence of its CDSs matched up with the gene *MdMYB66* in the apple genome, according to our last search. To explore the evolutionary relationship of *MdMYB66* protein, we constructed a phylogenetic tree of *MdMYB66* protein sequences and MYB homologous protein sequences from other species. The result showed that *MdMYB66* and PdMYB66 had high homology (Figure 7A). In addition, the *MdMYB66* protein and other plant MYB proteins were analyzed by multi-sequence alignment and structural, functional domain analysis using DNAMAN 8.0 software. The result showed that *MdMYB66* has two MYB-like DNA-binding domains (alignment region 9–56 and alignment region 62–107). That is, *MdMYB66* contains R2 and R3 MYB repeat domains belonging to the typical R2R3-MYB type of the MYB family (Figure 7B).

### 2.8. Tissue-Specific Expression Analysis and Subcellular Localization of MdMYB66

The result of tissue-specific expression pattern analysis of *MdMYB66* showed that the expression level of *MdMYB66* was the highest in fruit, followed by flowers, and the expression level was relatively lower in roots, stems, and leaves, which further confirmed that *the *MDMYB66** transcription factor may play a key role in fruit development and maturation (Figure 7C). Subcellular localization analysis showed that GFP labeled *MdMYB66* protein fluoresce was observed in the nucleus, but GFP signal was observed in the entire cell of 35S: GFP control, indicating that *MdMYB66* was localized in the nucleus and exerted its role primarily in the nucleus (Figure 7D).

### 2.9. Functional Analysis of MdMYB66 by Overexpression in Transgenic Materials 

To study the biological function of *MdMYB66* in the regulation of anthocyanin biosynthesis, an overexpression vector was constructed and transformed into apple skins by transient agroinfiltration in the ‘Golden Delicious’ apple. The results showed that the red color phenotype of overexpressed *MdMYB66* was obvious, while the control showed no color change in the apple skins (Figure 8A). Then, anthocyanin content was measured in the colored parts, as shown in Figure 8B. The anthocyanin extract of apple fruit skin was colorless after injection with the Empty vector as the control, while the anthocyanin extract of apple fruit skin with overexpression of *MdMYB66* was pink. The anthocyanin content of apple fruit skin injected with the Empty vector as control was 17 μg/g FW, while the anthocyanin content of apple fruit with overexpression of *MdMYB66* was significantly higher than that of the control, with the value of 190 μg/g FW, a 10-fold increase compared with the control. In addition, overexpression of *MdMYB66* significantly increased the expression levels of *MdMYB66* and structural genes *CHI*, *CHS*, *F3H*, *ANS*, *DFR*, and *UFGT* related to anthocyanin synthesis pathway (Figure 8C,D).

The function of *MdMYB66* was also studied in stably transformed ‘Orin’ apple callus by the agrobacterium-mediated transformation. The results showed that the callus of *MdMYB66*-OE gradually turned red, but there was no color change in the control callus (WT) (Figure 9A). In addition, there was a distinct change in the color phenotype of anthocyanin extract. The content of anthocyanin in the callus of *MdMYB66*-OE was significantly higher than that of WT (Figure 9B). The relative expression level of *MdMYB66* was significantly increased compared with WT (Figure 9C). Moreover, the relative expression levels of structural genes *CHI*, *CHS*, *F3H*, *ANS*, *DFR*, and *UFGT* related to the anthocyanin synthesis pathway were significantly increased compared with the control (Figure 10).

### 2.10. Y1H and LUC Test Demonstrated That MdMYB66 Combined with MdF3H Promoter Promoted Anthocyanin Synthesis

Transcriptome analysis showed that the expression level of *MdF3H* was significantly up-regulated in the mutant. To further characterize the function of *MdMYB66*, a 2000 bp region upstream of the translation start site in the *MdF3H* (the putative promoter sequence) was cloned and analyzed through the PlantCARE (http://bioinformatics.psb.ugent.be/webtools/plantcare/html/, accessed on 15 February 2023) program. Multiple MYB-binding elements and light-responsive cis-acting elements (G-boxes) were detected (Figure 11A). To determine whether *MdMYB66* could interact with *MdF3H*, a yeast one-hybrid (Y1H) assay was performed. The results showed that when the concentration of 3-AT reached 100 mM, the yeast strains containing the co-expression of *MdMYB66*-AD and proMdF3H-pHIS2 could grow normally on SD/-Trp/-Leu/-His plate, but no growth was observed on the control plate, indicating that *MdMYB66* could bind to the promoter of *MdF3H* (Figure 11B).

*MdMYB66* transcription factor was inserted into the pGreenII 62-SK vector, and the promoter sequence of *MdF3H* was inserted into the pGreenII 0800-LUC vector, and then tobacco leaves were co-injected and detected by a live fluorescence imaging system and luciferase activity. The results showed that the co-expression of 35S:*MdMYB66* + pF3H: LUC showed stronger luciferase luminescence signal intensity than the control group, indicating that *MdMYB66* positively regulated the expression of the *MdF3H* gene (Figure 11C). The results of the dual luciferase activity test showed that *MdMYB66* activated *MdF3H* 3.8 times (Figure 11D). These results indicate that *MdMYB66* could promote the expression of *MdF3H* and then affect the content of anthocyanins.

### 2.11. Modulation Reconstruction between Anthocyanin and Lignin Accumulation in OS and RM 

On the basis of the above results, a model for the mechanism by which the *MdMYB66*-*MdF3H* module regulates the balance between anthocyanin and lignin accumulation in OS and RM (Figure 12). In the early stages of fruit growth and development, *MdPAL* and *Md4CL* are up-regulated in the phenylpropanoid pathway. *MdF3H* is up-regulated in the anthocyanin biosynthesis, while the related lignin biosynthesis genes of *MdCCR* and *MdCAD* are downregulated. Notably, substrate competition exists between anthocyanin and lignin biosynthesis. *MdMYB66* expression is up-regulated and subsequently binds the promoter of the target gene *MdF3H*, resulting in the upregulation of the *MdF3H* expression and anthocyanin production. On the other hand, the upregulation of *MdMYB66* expression may weaken the inhibition of the transcription of lignin biosynthesis genes *MdCCR* and *MdCAD* and lignin accumulation.

## 3. Discussion

Anthocyanins are water-soluble flavonoids that widely exist and are distributed in plants, endowing different colors to plant tissues and organs. Meanwhile, anthocyanins are important secondary metabolites, and many types of anthocyanins have been identified in plants, of which pelargonidin (Pg), cyanidin (Cy), delphinidin (Dp), peonidin (Pn), petunidin (Pt) and malvidin (Mv) are commonly found in plants. Previous studies have shown that the skin of apples has five anthocyanin components [18,19]. Among these anthocyanins, the content of cyanidin-3-galactoside is the highest in apple skin [18]. In our study, the concentration of cyanidin-3-galactoside in RM was also at a relatively high level compared with OS, accounting for about 35% and 65% at early and later stages, respectively. The pigments of cyanidin-based anthocyanins, especially cyanidin-3-O-glucoside, cyanidin-3-O-galactoside, and cyanidin-3-O-xyloside, contributed to the red coloration of RM at early growth and developmental stage. Although OS also contains similar kinds of cyanidin pigments, their contents may be too low to be responsible for green coloration. In addition, this study also found that the content of peonidin-based anthocyanin pigment was significantly increased during the fruit ripening stage. Therefore, we inferred that the dark red color skin of RM was the result of the combined effect of cyanidin and peonidin.

The anthocyanin biosynthesis pathway and the key genes have been extensively studied in many plants [20]. The phenylpropanoid metabolism pathway can produce anthocyanin and lignin, which play a pivotal role in plant growth and development [21]. Flavonoids are also produced via the phenylpropanoid metabolism pathway and are indispensable secondary metabolites in plants [22]. By transcriptome analysis, we found that the expression levels of upstream two *4CL* genes were relatively higher in RM than in OS at an early stage. The gene of *4CL* catalyzed in the general phenylpropanoid pathway is one of the key genes involved in the synthesis of lignin, flavonoid, and other secondary metabolites. According to the different pathways involved in the synthesis of substances, *4CL* can be divided into two categories, among which class Ⅰ is involved in the synthesis of lignin monomer, while class Ⅱ is mainly involved in flavonoid synthesis [23]. Furthermore, the expression of the *F3H* gene was also relatively high in RM but low in OS. In general, the downstream of the phenylpropanoid pathway contains multiple metabolic branches, among which the flavonoid pathway and lignin pathway are the two main branches. The flavonoid pathway produces the largest group of polyphenol metabolites, a total of more than 6000 metabolites [24]. The *F3H* gene is also a key functional gene regulating the accumulation of flavonoid metabolites in plants [25]. Therefore, in this study, we speculated that the structural gene *F3H* could have a positive effect on regulating the biosynthesis of anthocyanin in RM fruit skin. 

Lignin biosynthesis is another branch of the phenylpropanoid pathway, and its synthesis is also catalyzed by a series of structural genes, such as Hydroxycinnamoyl-CoA shikimate/quinate hydroxycinnamoyl transferase (*HCT*), Cinnamoyl-CoA reductase (*CCR*), and cinnamyl alcohol dehydrogenase (*CAD*). Previous studies showed that silencing the *HCT* gene inhibits lignin biosynthesis and leads to the redirection of metabolic flux to flavonoid biosynthesis [26]. Overexpressing miR7125 or inhibiting *CCR* transiently in apple fruit increased anthocyanin biosynthesis but reduced lignin production under light-induced conditions [27]. Furthermore, in Arabidopsis, down-regulating the expression of *CCR*, *CAD*, and *HCT* significantly reduced the content of lignin, and the plants were smaller with stunted development but increased the flavonoid content [26,28]. In the present study, compared with OS, the expression levels of *CCR* and *CAD*-related to lignin biosynthesis were relatively lower in RM. The results indicated that when the lignin biosynthesis was restrained, the metabolic flux had to be redistributed to the flavonoid biosynthesis pathway, thus showing a color difference between RM and OS.

In addition to structural gene regulation, both lignin and flavonoid synthesis are also regulated by MYB TFs in the growth and development of plants [29]. Various genes related to anthocyanin biosynthesis have been well-known in previous studies. For example, the well-studied TFs *MdMYB1*, *MdMYBA*, and *MdMYB10* have been validated to regulate the pigment of apple skin and pulp [30]. The biosynthesis of flavonoids and lignin is produced through the phenylpropanoid pathway. Studies have shown that the biosynthesis of flavonoids and lignin can be synergistically regulated through a specific transcriptional regulatory network. Overexpression of *AtMYB4* inhibited lignin and flavonoid synthesis and thus suppressed plant growth in Arabidopsis [31]. In bananas, overexpression of *MYB31* resulted in decreased lignin deposition in the secondary wall of vascular elements by down-regulating the genes involved in lignin synthesis [32]. Moreover, overexpression of *PtMYB6* up-regulated the genes related to flavonoid biosynthesis and down-regulated the genes related to lignin biosynthesis, thereby resulting in increased anthocyanins and proanthocyanins in transgenic poplars [30]. In the ‘Fuji’ apple bud mutant, overexpression of *MdMYB90-like* can induce the expression of early and late genes such as *MdCHS*, *MdCHI*, *MdANS*, and *MdUFGT* [33]. In our study, we further found that the expression level of *MdMYB66* was significantly up-regulated in RM. That is, on the one hand, transcription factor *MdMYB6*6 could increase the anthocyanin content by up-regulating the expression level of *MdF3H*. On the other hand, *MdMYB6*6 could decrease the lignin content by down-regulating the expression level of CCR and CAD. In conclusion, we generated evidence that *MdMYB66* regulates anthocyanin biosynthesis in the mutant. 

## 4. Materials and Methods

### 4.1. Plant Materials

A new red bud mutant (RM) from ‘Oregon Spur Ⅱ’ (OS) of the Red Delicious apple (*Malus domestica* Borkh.) was previously discovered on a branch at an apple orchard in Tianshui, Gansu Province, China, in 2017. The fruits of RM were always red from the young fruitlet stage to the maturity stage and, therefore, were readily distinguishable from the fruits of OS (Figure 1A). The phenotypes of the leaves and the phenological periods of RM and OS were not significantly different. After grafting, the bud mutant phenotype was stably inherited.

Fruit skin samples of ‘Oregon Spur Ⅱ’ and its red bud mutant were collected at 20, 60, 100, 120, 130, and 140 days after fluorescence (DAF), respectively. The skin and pulp of nine fruits were separated by approximately 1 mm of the skin tissue at each stage. All samples were collected with three biological replicates and then immediately frozen in liquid nitrogen and kept at −80 °C in a refrigerator for subsequent analysis. Next, 20, 60, 100, 120, 130, and 140 DAF specimens were collected for anthocyanin content analysis. Twenty DAF specimens were collected for RNA extraction and RNA-seq analysis. Twenty DAF and 130 DAF specimens were collected for metabolomic analysis. 

### 4.2. Determination of Anthocyanin Content

The total anthocyanin contents in the fruit skins of OS and RM were determined as described previously at six stages, with slight modifications [34]. In brief, 0.5 g skin tissue samples were added to 10 mL precooled 1% hydrochloric acid-methanol solution and homogenized with liquid nitrogen, and then transferred to a 20 mL scale test tube and extracted at 4 °C for 20 min under light. The filtrate was then filtered by a 0.2 µm polyethersulfone membrane (KrackelerScientific, Inc., Albany, NY, USA) and collected for later use. ATU-1900 double beam UV-visible spectrophotometer (Beijing Purkinje General Instrument Co., LTD, Beijing, China) was used, and the supernatant absorbance was determined to be 530 nm and 600 nm. The difference between 530 nm and 600 nm shows the relative anthocyanin content. Three replications were included per sample, and the data are shown as the means ± SD.

### 4.3. Metabolomic Profiling 

The stages of 20 DAF (S1) and 130 DAF (S2) were selected for Metabolomic analysis. S1 was the young fruitlet stage, and S2 was at the fruit ripening stage. Metabolomic analysis was performed by Wekemo Bioincloud Technology Co., Ltd. (Shenzhen, China) (https://www.bioincloud.tech/, accessed on 10 October 2022). In brief, 500 mg freeze-dried fruit skin samples were grounded and homogenized with liquid nitrogen into powder and extracted with 0.5 mL methanol/water/hydrochloric acid (500:500:1, *V*/*V*/*V*). Then, the extract was vortexed for 5 min and ultrasound for 5 min and centrifuged at 12,000 g under 4 °C for 3 min. The supernatants were collected and filtrated through a 0.22 μm membrane filter (Anpel, ANPEL Laboratory Technologies (Shanghai) Inc., Shanghai, China) before LC-MS/MS analysis. 

### 4.4. RNA Extraction, Quantification and Sequencing

Fruit skin samples at the 20 DAF developmental stage were selected for RNA-seq. RNA extraction, Library Construction, and transcriptome sequencing were performed by the Oxford Nanopore Technologies (ONT, Oxford, UK) platform at Biomarker Technology Company (Beijing, China). One μg total RNA was prepared for cDNA libraries using the cDNA-PCR Sequencing Kit (SQK-PCS109, Oxford Nanopore Technologies, Oxford, UK) protocol provided by Oxford Nanopore Technologies (ONT). Briefly, the template-switching activity of reverse transcriptases enriches full-length cDNAs and adds defined PCR adapters directly to both ends of the first-strand cDNA. And following cDNA PCR for 14 circles with LongAmp Tag (NEB, New England Biolabs, Inc., Ipswich, MA, USA). The PCR products were then subjected to ONT adaptor ligation using T4 DNA ligase (NEB). Agencourt XP beads were used for DNA purification according to ONT protocol. The final cDNA libraries were added to FLO-MIN109 flowcells and run on the PromethION platform. 

### 4.5. qRT-PCR

qRT-PCR analysis was performed using SYBR Green PCR Master Mix (Takara, Dalian, China) kit with a specific reaction system of 20 µL (2 × SYBR Green Master Mix 10 μL; cDNA 1 μL; ddH_2_O 7 μL; primer 2 μL). After the reaction solution was thoroughly mixed, the reaction was performed using the LightCycler^®^96 SW 1.1 real-time fluorescent quantitative PCR apparatus, and each sample was repeated three times. Reaction procedure (denaturation at 95 °C for 30 s; Denatured at 95 °C for 5 s; Annealing at 60 °C for 30 s; Extension at 95 °C for 15 s; 60 °C for 30 s; 95 °C for 15 s; 40 cycles). *MdGADPH* was used as the internal reference gene, and the relative expression level was calculated using the 2^−ΔΔCt^ method [35]. All primers are used in Appendix A.

### 4.6. Subcellular Localization 

The subcellular localization of *MdMYB66* was conducted as previously described [36]. The open reading frame (ORF) sequence of *MdMYB66* without the stop codon was cloned and inserted into the pCAMBIA1301: GFP expression vector. The fusion construct 35S:*MdMYB66*: GFP was transformed into *N. benthamiana* leaf epidermal cells. Fluorescence was detected by confocal microscopy after 2–3 days of transfection.

### 4.7. Construction of Plasmids and Genetic Transformation

The overexpression and interference plasmids were constructed as previously described [37]. The full-length CDS of *MdMYB66* was cloned into a pCAMBIA1301 vector to generate a 35S:*MdMYB66* construct. The recombinant plasmid was introduced into Agrobacterium strain GV3101 and transformed into ‘Orin’ apple callus and ‘Golden Delicious’ fruit skins. Transgenic callus was screened based on kanamycin resistance. Transgenic callus and WT callus were grown at 24 °C under dark conditions and subcultured every 15 days on media supplemented with kanamycin. Three lines of transgenic callus were harvested, transferred to new plates, and cultured for 5 days under light conditions. Anthocyanin contents and the expression of genes related to anthocyanin biosynthesis were analyzed in both transgenic and WT callus. All primers are used in Appendix A.

### 4.8. Yeast One-Hybrid Assay

Y1H (yeast one-hybrid) assays were conducted as previously described [37]. *MdMYB66* coding sequence (CDSs) was cloned into pGADT7 vector to generate *MDMYB66*-AD recombinant vector and the promoter sequence of *MdF3H* was inserted into pHIS2 vector to generate proMdF3H-pHIS2 recombinant vector for Y1H detection. Then, 3-AT was used to screen suitable concentrations to inhibit the self-activation of the pHIS2 vector itself. All primers are used in Appendix A.

### 4.9. LUC Assay

The LUC analysis was conducted as previously described [37]. The 62SK-MdMYB66 effectors (35S:MdMYB90-like cloned into pGreenII62-SK vector) and reporter constructs (the promoter fragments of *MdF3H* cloned into the pGreenII 0800-LUC vectors) were transformed into *A. tumefaciens* GV3101(p-soup). The bacteria were mixed and co-injected into tobacco leaves and cultured for 1 day under dark conditions. A living fluorescence imager was used to observe the fluorescence of the tobacco leaves, which were also sampled to measure LUC/REN activity. All primers are used in Appendix A.

### 4.10. Statistical Analysis 

SPSS 20.0 software was used to analyze statistical significance by one-way analysis of variance (ANOVA). Data are presented as means ± standard deviations (SD) of three independent biological replicates. The significant differences among samples were analyzed with Duncan’s test. Differences with *p* < 0.05 were statistically significant. Figures were constructed using Origin version 9.6. 

## 5. Conclusions

In summary, this study found that the anthocyanin content in RM was significantly higher than that in OS during fruit development. Through widely targeted metabolomic, we found that the contents of cyanidin anthocyanins, including cyanidin-3-O-galactoside and cyanidin-3-O-arabinoside, were significantly up-regulated in the fruit skin of RM at the early stage. Transcriptomic analysis showed that the expression level of *F3H* involved in anthocyanin biosynthesis was significantly higher in RM compared with OS, while the expression levels of *CCR* and *CAD* involved in lignin biosynthesis were significantly down-regulated. A key transcription factor, *MdMYB66*, which is involved in anthocyanin regulation in the mutant, was screened out. Then, the function of *MdMYB66* was verified by transforming apple fruit and callus. The results showed that overexpression of *MdMYB66* significantly increased the contents of anthocyanin in apple skin and callus and significantly increased the expression level of genes related to anthocyanin synthesis. In addition, the Y1H and LUC tests also verified that *MdMYB66* could bind to the promoter of the downstream target gene *MdF3H* and affect anthocyanin contents. These studies provide a more detailed understanding of the anthocyanin regulatory network during apple fruit development and a new perspective for studying anthocyanin biosynthesis in anthocyanin-deficient plants.

## Figures and Tables

**Figure 1 ijms-24-16871-f001:**
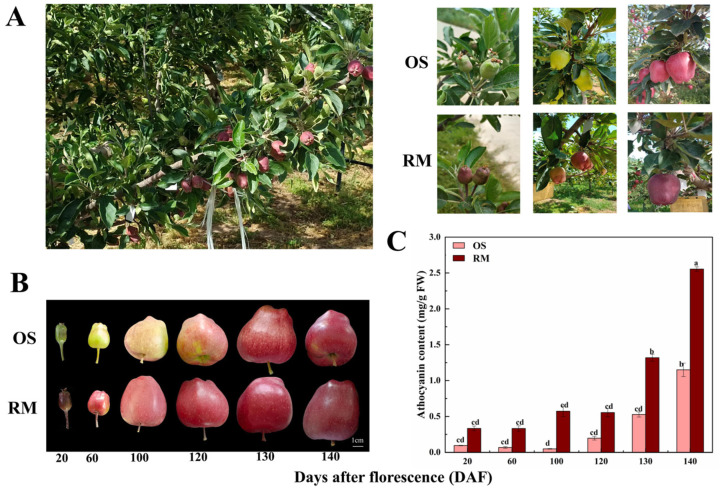
Phenotype and anthocyanin content of OS and RM. (**A**) The fruit tree and phenotype of OS and RM at different growth and development stages. (**B**) Fruit phenotype of OS and RM at six developmental stages. (**C**) Contents of total anthocyanins. Values presented are the mean ± SD (Standard deviation) from three biological replicates. Different letters indicate significant differences among treatments in the same period using Duncan’s test (*p* < 0.05).

**Figure 2 ijms-24-16871-f002:**
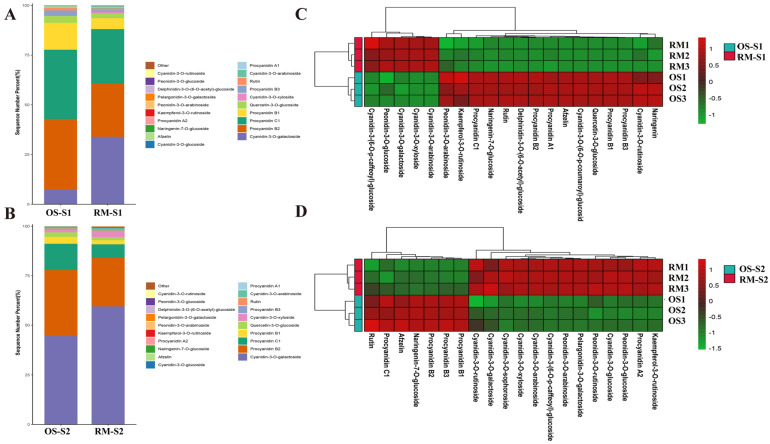
Variations in anthocyanin composition and concentration in OS and RM. (**A**,**B**) Comparison of the top 20 relative abundances of anthocyanins at S1 (**A**), S2 (**B**) stage in OS and RM. (**C**,**D**) Hierarchical cluster heatmap analysis based on the relative content of anthocyanins.

**Figure 3 ijms-24-16871-f003:**
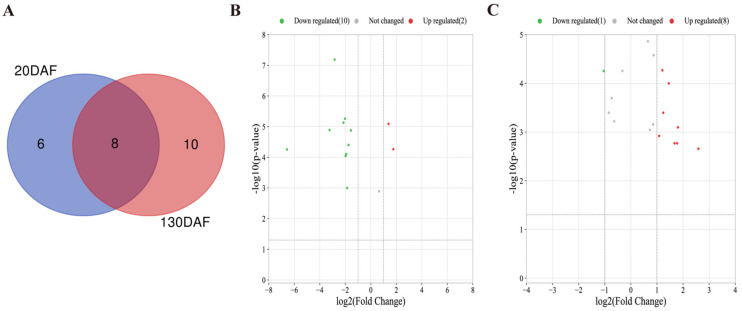
Differentially accumulating anthocyanins (DAAs) between OS and RM at 20 DAF and 130 DAF. (**A**) Venn diagram of differential metabolites. (**B**,**C**) Volcano plot of differential metabolites. Each point represents a metabolite, the X-axis is log_2_Fold Change, and the Y-axis is the *p*-value of the *t*-test. log_2_Fold Change is positive, indicating up-regulated. log_2_Fold Change is negative, indicating down-regulated.

**Figure 4 ijms-24-16871-f004:**
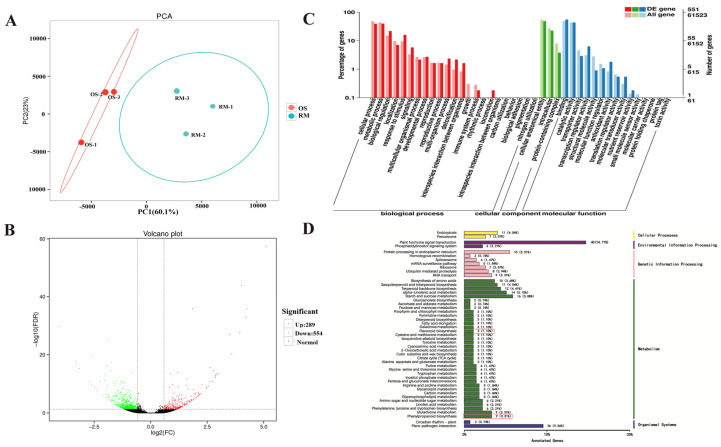
Transcriptome profiling of OS and RM. (**A**) Principal component analysis (PCA) of the two skin samples. (**B**) Volcano plot of DEGs. The green dots indicate the down-regulated expressed genes, and the red dots indicate the up-regulated expressed genes. (**C**) GO annotation classification and enrichment analysis of DEGs. The Y-axis represents the number of DEGs enriched in each GO classification. (**D**) KEGG pathway enrichment analyses of DEGs. The red boxes indicate the metabolic pathways associated with anthocyanin synthesis and the number of DEGs.

**Figure 5 ijms-24-16871-f005:**
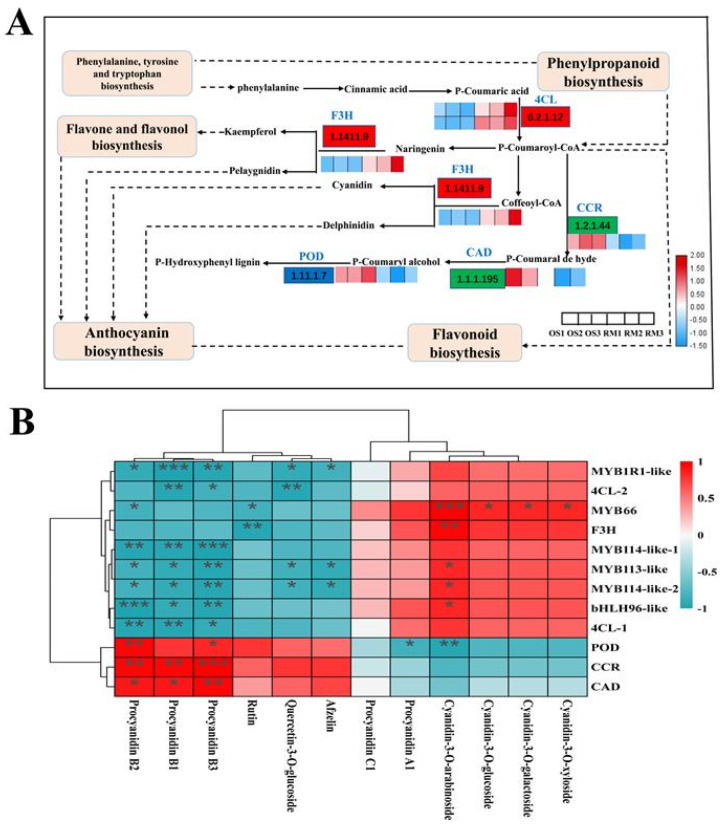
DEGs analysis. (**A**) Differential expression of structural genes involved in the anthocyanin biosynthesis pathway during the coloration of OS and RM (from left to right: the expression level of OS1, OS2, OS3, RM1, RM2, RM3, respectively). The abbreviations of enzymes are as follows: 4CL (4-coumarate-CoA ligase); F3H (naringenin 3-dioxygenase); CCR (cinnamoyl-CoA reductase); CAD (cinnamyl-alcohol dehydrogenase); POD (peroxidase). (**B**) Correlation heatmap analysis between DAAs and DEGs in OS and RM. The charts in blue and red represent the down- and up-regulated, respectively. Statistical significance of difference was expressed by *p* < 0.05 (*), *p* < 0.01 (**) and *p* < 0.001 (***), respectively.

**Figure 6 ijms-24-16871-f006:**
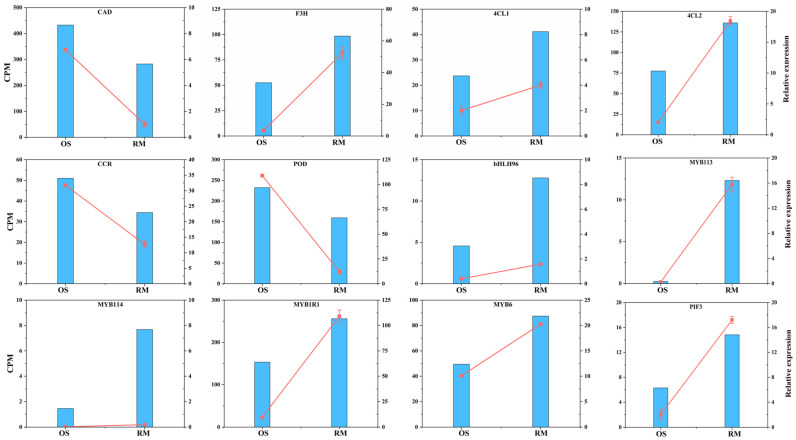
qRT-PCR validation of 12 selected DEGs in OS and RM. The left y-axis indicates the corresponding expression data from RNA-seq (blue histogram). The right y-axis shows the relative gene expression level measured by qRT-PCR (red lines). Bars represent the standard error (SE; *n* = 3).

**Figure 7 ijms-24-16871-f007:**
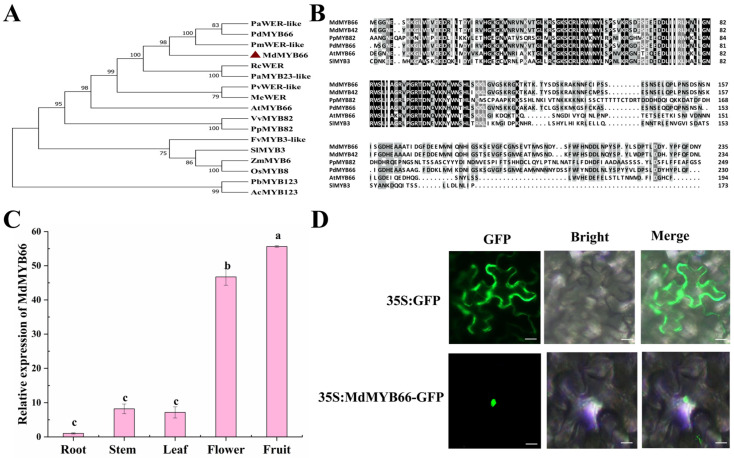
Bioinformatics analysis of *MdMYB66*. (**A**) Phylogenetic tree analysis of *MdMYB66*. (**B**) Multi-sequence alignment analysis of *MdMYB66*. R2 and R3 are MYB-like DNA-binding domains. (**C**) Tissue expression analysis of *MdMYB66*. Data are shown as mean ± standard deviation, *n* = 3. Bars with different letters are significantly different at *p* < 0.05. (**D**) Subcellular localization analysis of *MdMYB66*.

**Figure 8 ijms-24-16871-f008:**
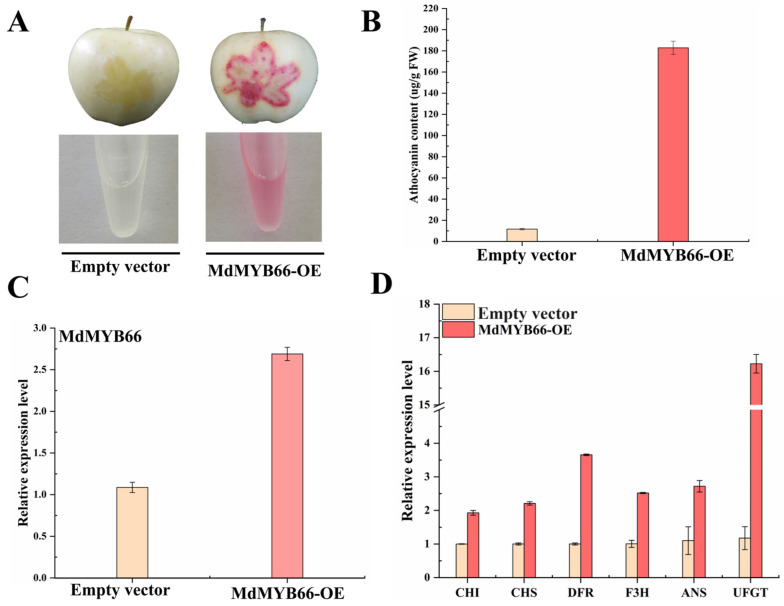
The instantaneous infection of *MdMYB66* on apple fruit promoted the synthesis of anthocyanin. (**A**) The phenotype of apple fruit and anthocyanin extract after instantaneous injection of *MdMYB66*. (**B**) Determination of anthocyanin content in apple injected with *MdMYB66*. (**C**) The expression level of *MdMYB66* in apple fruit that was immediately infected by *MdMYB66*. (**D**) Expression levels of structural genes related to anthocyanin biosynthesis in apple fruits infected by *MdMYB66*.

**Figure 9 ijms-24-16871-f009:**
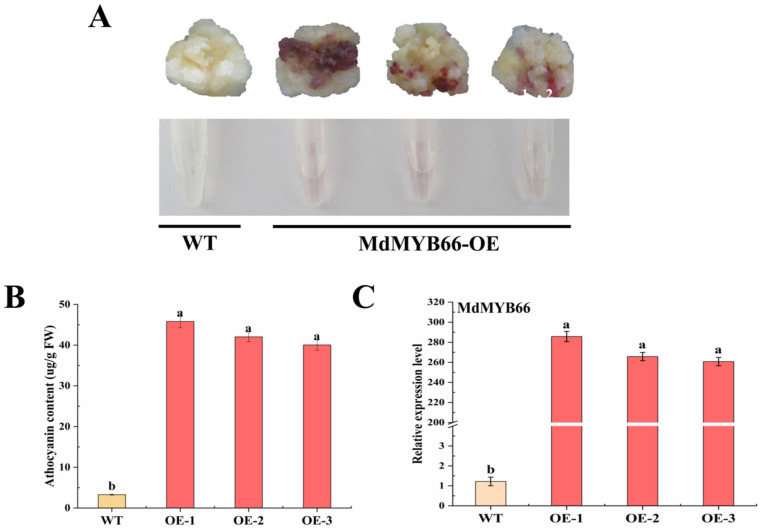
Functional verification of overexpression of *MdMYB66* in ‘Orin’ apple callus. (**A**) Color phenotypes and anthocyanin extract of wild type (WT) and overexpressed *MdMYB66* callus. (**B**) The contents of anthocyanin in wild-type (WT) and overexpressed *MdMYB66* callus. (**C**) Expression levels of *MdMYB66* in wild-type (WT) and overexpressed callus tissues. Data are shown as mean ± standard deviation, *n* = 3. Bars with different letters are significantly different at *p* < 0.05.

**Figure 10 ijms-24-16871-f010:**
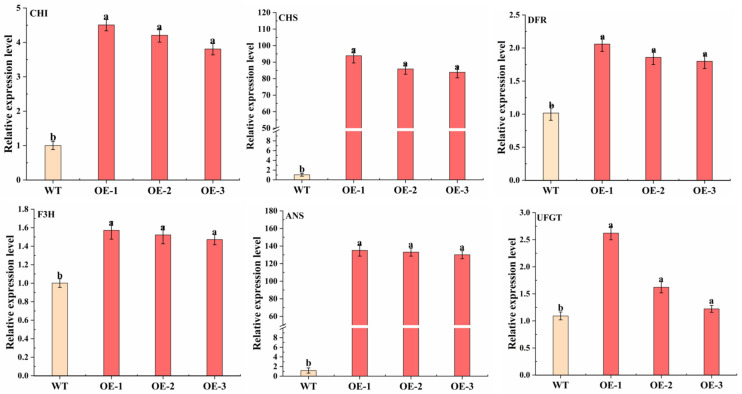
Expression levels of structural genes associated with anthocyanin biosynthesis in the callus of overexpression *MdMYB66*. Data are shown as mean ± standard deviation, *n* = 3. Bars with different letters are significantly different at *p* < 0.05.

**Figure 11 ijms-24-16871-f011:**
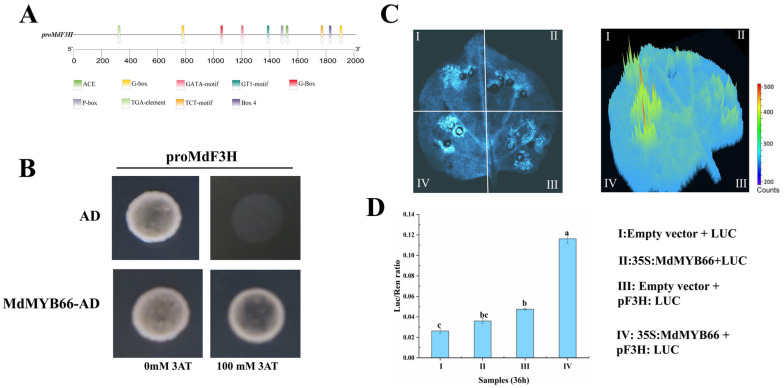
Y1H and LUC verified that *MdMYB66* could bind the promoter of *MdF3H*. (**A**) Analysis of cis-acting elements of the *MdF3H* promoter. (**B**) Y1H confirmed the binding of *MdMYB66* and the promoter of *MdF3H*. AD as control. (**C**) 2D and 3D images of LUC after transient penetration of tobacco leaves. (**D**) The ratio of LUC to REN activity. The values represent the mean ± standard deviation of the three experiments. Lowercase letters indicate a significant difference relative to the control vector (*p* < 0.05).

**Figure 12 ijms-24-16871-f012:**
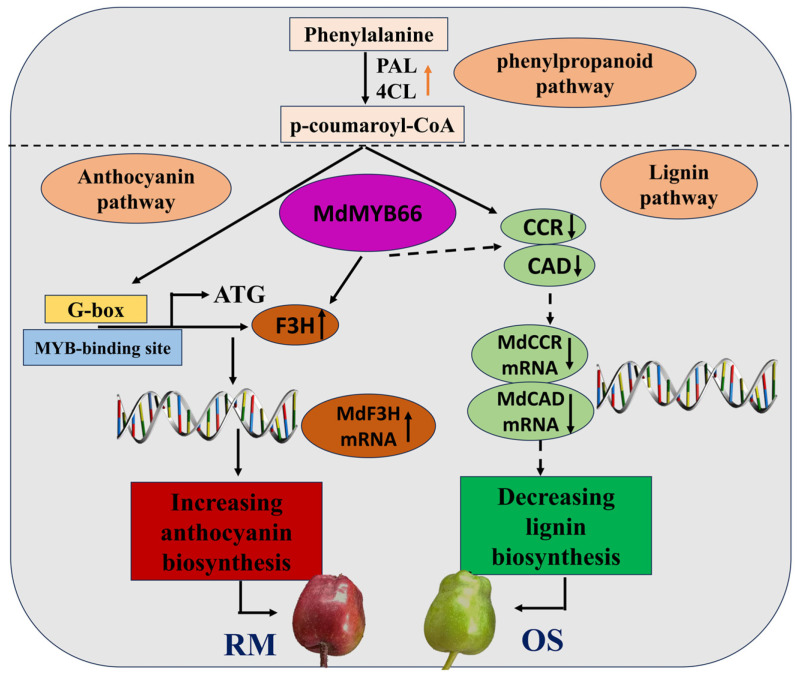
A model for the mechanism by which the *MdMYB66*-*MdF3H* module regulates the balance between anthocyanin and lignin accumulation in OS and RM.

**Table 1 ijms-24-16871-t001:** Summary of RNA-Seq data and mapping.

Sample	ReadNum	Clean Reads	BaseNum	N50	Mean Qscore	Full-Length Reads	Full-Length Percentage (FL%)
OS-1	6,213,112	6,017,549	6,875,665,663	1240	Q12	5,293,191	0.8796
OS-2	7,869,345	7,595,478	9,274,380,168	1326	Q12	6,647,381	0.8752
OS-3	5,342,582	5,175,741	6,172,885,434	1305	Q12	4,583,013	0.8855
RM-1	5,007,883	4,861,294	5,797,835,509	1313	Q12	4,351,298	0.8951
RM-2	5,562,398	5,382,443	6,783,270,143	1421	Q12	4,717,954	0.8765
RM-3	6,756,504	6,523,122	7,573,741,055	1252	Q12	5,795,413	0.8884

## Data Availability

All data generated or analyzed during this study are included in this published article and its Appendix A.

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
