# Peer review of "MdMYB66 Is Associated with Anthocyanin Biosynthesis via the Activation of the MdF3H Promoter in the Fruit Skin of an Apple Bud Mutant"

_ijms, 2023, doi:10.3390/ijms242316871_

Round 1
Reviewer 1 Report
Comments and Suggestions for Authors
This research article deals with the anthocyanin content of a new bud mutant of a cultivar of apple (Malus domestica Borkh.). The coloring mechanism of ‘Oregon Spur â…¡’ and its red mutant at different is studied at developmental stages, transcriptomic integrated with metabolomic is performed, a key regulatory gene, MdMYB66, is uncovered as a novel R2R3 type MYB transcription factor. The mechanism of anthocyanin biosynthesis is discussed. The experimental results are, well presented and the interpretations are convincing. They can also help in the future in developing bioactive compounds and cultivating anthocyanin-rich red apple varieties
Comments
The importance of anthocyanin for human health should be described with a little bit more detail |line 39|.
Comments on the Quality of English LanguageModerate editing of English language required
For example
Apple (Malus domestica Borkh.) is very popular and economical fruits all over the world due to its .....
should be
Apple (Malus domestica Borkh.) is a very popular and economical fruit all over the world due to its .....
Reviewer 2 Report
Comments and Suggestions for Authors
Excellent manuscript by Huang et al.
Minor changes I suggest are the following.
Line 30- "People's Daily" to "people's daily"
Line 99- Full name needed for DAF. I know it was mentioned in the materials and methods part, but the results section comes first.
Figure 1 captions have two (B). Line 114 should be changed to (C).
Line 160- p(italic)-value
Figure 3 caption & Line 170- at 20(space)DAF
Line 221-222- Figure 7B is mentioned prior to other figures like 5 or 6. I know it is hard to be in order, but the journal might want to do so.
Figure 5 B- Is the correlation combined for both maturity stages + WT & mutant? It's not crystal clear.
Line 453- 4(space)degree celsius like line 444
Line 520- Italic for A. tumefaciens
Overall, it is a very well-written manuscript.
